# Can Lemborexant for Insomnia Prevent Delirium in High-Risk Patients with Pancreato-Biliary Disease after Endoscopic Procedures under Deep Sedation?

**DOI:** 10.3390/jcm12010297

**Published:** 2022-12-30

**Authors:** Takeshi Ogura, Saori Ueno, Atsushi Okuda, Nobu Nishioka, Akira Miyano, Yoshitaro Yamamoto, Kimi Bessho, Mitsuki Tomita, Nobuhiro Hattori, Junichi Nakamura, Hiroki Nishikawa

**Affiliations:** 1Endoscopy Center, Osaka Medical and Pharmaceutical University, Osaka 569-8686, Japan; 22nd Department of Internal Medicine, Osaka Medical and Pharmaceutical University, Osaka 569-8686, Japan

**Keywords:** ERCP, EUS-guided drainage, lemborexant, insomnia, endoscopy

## Abstract

**Background and aim**: Pancreato-biliary patients who undergo endoscopic procedures have high potential risk of delirium. Although benzodiazepine has traditionally been used to treat insomnia, this drug might increase delirium. Lemborexant may be useful for patients with insomnia, without worsening delirium, although there is no evidence for high-risk patients with pancreato-biliary disease. The aim of this pilot study was to evaluate the safety and efficacy of lemborexant for insomnia and the frequency of delirium after endoscopic procedures under deep sedation in patients with pancreato-biliary disease. **Method:** This retrospective study included consecutive patients who were administered lemborexant after endoscopic procedures for pancreato-biliary disease between September 2020 and June 2022. The primary outcome of this study was evaluation of the safety and efficacy of lemborexant for insomnia. Frequency of delirium was the secondary outcome. **Result:** In total, 64 patients who had the complication of insomnia after an endoscopic procedure were included in the study. Risk factors for delirium were advanced age (n = 36, 56.3%), dementia (n = 10, 15.6%), and regular alcohol use (n = 13, 20.3%), as well as the sedatives midazolam and pentazocine that were administered to all patients at the time of the endoscopic procedure. Successful asleep was achieved by 61/64 patients (95.3%). No fall event was observed during the night following the procedure in any patient. However, mild consciousness transformation was observed in one patient. **Conclusions:** In conclusion, lemborexant use may be effective and safe for use after endoscopic procedures in pancreato-biliary patients, without increasing the risk of delirium.

## 1. Introduction

Delirium can be characterized by the acute onset of cerebral dysfunction with a fluctuation or change in mental situation, inattention, and either disorganized thinking or altered level of consciousness [1]. A recent meta-analysis reported that delirium occurs in 23% of inpatients [2]. Delirium, which causes distress to patients and caregivers, has been associated with increasing morbidity and mortality, and is a major burden to healthcare services regarding costs [3]. Total cost estimates attributable to delirium have been reported to range from USD 16,303 to USD 64,421 per patient, implying that the national burden of delirium on the healthcare system ranges from USD 38 billion to USD 152 billion annually [3]. Therefore, in clinical practice, we must pay attention to reducing all factors associated with delirium.

Pancreato-biliary diseases such as pancreatic cancer, bile duct cancer, alcoholic chronic pancreatitis, or cholangitis are usually needed to be hospitalized. Various invasive endoscopic procedures such as endoscopic retrograde cholangiopancreatography (ERCP) or endoscopic ultrasound-guided biliary drainage (EUS-BD) may be attempted for such cases [4,5,6]. These procedures usually require deep sedation, which must be performed safely. Midazolam is commonly used as a sedation drug because of its limited effect on hemodynamics and short half-life [7]. However, several studies have shown that midazolam has risk for delirium [8,9,10]. Shi et al. conducted an observational study regarding the association of exposure to midazolam within 24 h prior to delirium assessment and the risk of delirium [9]. They reported that 28.28% of 4808 patients who were administered midazolam experienced delirium. Compared with patients who were not administered midazolam, those treated with midazolam exhibited a significant difference (odds ratio [OR] 2.54; 95% CI 2.31–2.76; *p* < 0.001). According to a recent network meta-analysis [11], dexmedetomidine was associated with a lower incidence of delirium (OR 0.44; 95% CI 0.30–0.64) compared with placebo, whereas midazolam was associated with a higher incidence of delirium (OR 2.62; 95% CI 1.07–6.43). This finding may indicate the importance of paying attention to the occurrence of delirium after endoscopic procedures when midazolam is used for sedation. When the procedures are performed in the daytime, some patients experience insomnia after waking up from sedation, and sleeping drugs may be needed to counter this effect. In addition, many Japanese endoscopists use midazolam and pentazocine during sedation, which carry the risk of postoperative delirium; therefore, selection of the sedative drug is clinically important. Although benzodiazepine has traditionally been used to treat insomnia, this drug might increase delirium because it decreases rapid eye movement sleep (REMS) [12,13]. Therefore, benzodiazepines should be avoided for patients at high risk of delirium, including those with pancreato-biliary disease. Lemborexant, a dual orexin receptor antagonist that acts on orexin 1 and 2 receptors, has recently been approved in several countries, including Japan. A global phase 3 trial [14] has confirmed the clinical safety and efficacy of lemborexant for older adult patients with insomnia, without decreasing the amount of REMS [15] or worsening delirium [14]. Indeed, suvorexant, which is another dual orexin receptor antagonist, may have evidence regarding preventing delirium. However, there is no evidence regarding this hypothesis of lemborexant for high-risk patients with pancreato-biliary disease.

The aim of this pilot study was to evaluate the safety and efficacy of lemborexant for insomnia and the frequency of delirium after endoscopic procedures under deep sedation in patients with pancreato-biliary disease.

## 2. Patients and Methods

This retrospective study included consecutive patients who were administered lemborexant after endoscopic procedures for pancreato-biliary disease between September 2020 and June 2022. All study protocols were approved by the institutional review board of Osaka Medical and Pharmaceutical University Hospital (IRB No. 2022-154). The study protocol conformed to the ethical guidelines of the 1975 Declaration of Helsinki as reflected in the a priori approval given by the human research committee at Osaka Medical and Pharmaceutical University. The inclusion criteria were (1) inpatient status, (2) complicated by pancreato-biliary disease, (3) requiring endoscopic procedures with deep sedation, (4) meeting the DSM-5 criteria for insomnia [16], and (5) age ≥18 years. Additionally, excluded were patients who experienced delirium during or immediately after the endoscopic procedure.

### 2.1. Sedation and Endoscopic Procedures

Endoscopic procedures were performed under deep sedation. Depth of sedation was evaluated according to the American Society of Anesthesiologists [13] or Ramsay sedation scores (4 or 5) [17]. Sedation was performed by experienced endoscopists. As sedation drugs, midazolam and pentazocine were administrated. Endoscopists initiated the sedation with 3–5 mg each of midazolam and pentazocine. Then, the depth of sedation was evaluated two or three minutes thereafter. In the case that deep sedation was not achieved, an additional dose of each drug was given as appropriate, and as required during the endoscopic procedure.

Endoscopic procedures were performed after obtaining deep sedation. Technical tips for ERCP were as follows. The duodenoscope was inserted into the duodenum and pancreato-biliary cannulation was performed. After successful cannulation, a 0.025-inch guidewire was deployed, and endoscopic sphincterotomy was performed if necessary. The treatment procedure was then performed (stent deployment, stone removal using a balloon or basket catheter, or electrohydraulic lithotripsy under cholangioscopy). Technical tips for EUS-guided biliary drainage were as follows. First, the echoendoscope was inserted into the stomach or duodenum, and the target site of interest was identified. The target was punctured under EUS guidance using a 19G needle, and guidewire deployment. After tract dilation, a plastic or covered self-expandable metal stent was deployed from the target organ to the intestine.

### 2.2. Definitions and Statistical Analysis

The primary outcome of this study was evaluation of the safety and efficacy of lemborexant for insomnia. Frequency of delirium during 48 h after Lemborexant administration was the secondary outcome. According to the DMS-5 criteria, insomnia is divided into the four subtypes of ‘difficulty initiating sleep’, ‘difficulty maintaining sleep’, ‘early-morning awakening’, and ‘non-restorative sleep’, which are defined as follows. ‘Difficulty initiating sleep’ as taking longer than 30 min to fall asleep, ‘difficulty in maintaining sleep’ as repeated waking after falling asleep, ‘early-morning awakening’ as waking ≥30 min before the scheduled wake time with total sleep time of ≤6.5 h, and ‘non-restorative sleep’ as the absence of rest despite having sufficient sleep. Delirium was diagnosed using the Confusion Assessment Method (CAM) [15]. The CAM comprises the following four features: acute onset or fluctuating course (1), inattention (2), disorganized thinking (3), and altered level of consciousness (4). Delirium was diagnosed by CAM in the presence of features (1) and (2), and either (3) or (4). These data were collected retrospectively from the patients’ medical records by the medical records. Based on previous studies, the following were considered high risk factors for delirium: advanced age (>70 years), regular alcohol use, dementia, drugs (e.g., H2 blocker, hypnotics, and antipsychotics), or past history of delirium [18,19]. Regular alcohol use was defined as a daily intake of >60 g of ethanol.

Hypoxemia was defined if continuous SpO_2_ ≤ 90% for at least 15 s, and SpO_2_ ≤ 85% that continued for >15 s was considered severe hypoxemia. Additionally, apnea was defined as etCO_2_ or a respiratory rate of 0 for at least 30 s [20]. Procedural time was measured from endoscope insertion to removal. The physical condition of the patient was evaluated according to the Eastern Cooperative Oncology Group performance status (ECOG-PS) [21]. Finally, descriptive statistics are presented as the mean ± standard deviation (SD) or as the median and range for continuous variables, and as the frequency for categorical variables.

## 3. Results

Table 1 lists the patients’ characteristics. In total, 64 patients (median age, 71 years; 47 males, 17 females) who had the complication of insomnia after an endoscopic procedure were included in the study. The primary disease was pancreatic tumor (n = 15), bile duct cancer (n = 10), chronic pancreatitis (n = 13), bile duct stone (n = 11), hepaticojejunostomy stricture (n = 7), and other (n = 8). Median body mass index was 21.8 kg/m^2^, and e-GFR was 71 mL/min. Mean PS was 0.68. Mean values of laboratory data before administration of lemborexant were as follows: total bilirubin, 2.82 ± 5.72 mg/dL; C-reactive protein, 3.10 ± 5.18 mg/L; aspartate aminotransferase, 55.6 ± 75.8 IU/L; alanine aminotransferase, 56.1 ± 98.3 IU/L. Eight patients (12.5%) had a past history of insomnia.

Risk factors for delirium were advanced age (n = 36, 56.3%), dementia (n = 10, 15.6%), and regular alcohol use (n = 13, 20.3%) (Table 2), as well as the sedatives midazolam and pentazocine that were administered to all patients at the time of the endoscopic procedure. Additional drugs were administered in some patients: H2 blocker (n = 28, 43.8%), hypnotics (n = 8, 12.5%), and antipsychotics (n = 1, 1.5%).

Table 3 lists the details of the endoscopic procedures. Stent placement under ERCP guidance was performed most frequently (n = 33), followed by stone removal (n = 11), EUS-guided biliary drainage (n = 11), EUS-FNA and biliary drainage (n = 4), EUS-FNA (n = 3), and EUS and biliary drainage (n = 2). The mean dose of midazolam was 6.81 mg, and that of pentazocine was 8.10 mg. Mean procedure time was 36.7 min. Several adverse events were observed during the endoscopic procedures, but were not severe.

Table 4 lists the sleep outcomes following administration of 5 mg of lemborexant. Difficulty initiating sleep was the most common subtype of insomnia (n = 52, 81.2%). Successful asleep was achieved by 61/64 patients (95.3%), and mid-awakening after sleep was observed in 3 patients. These 3 patients slept for several hours with no additional medication. Among 58 patients who achieved successful sleep with lemborexant, 57 successfully woke at the scheduled time. No fall event was observed at the first night in any patient. However, mild consciousness transformation was observed in one patient at the first night. Finally, during 72 h observation, any events associated with delirium was not observed. As adverse events, liver damage due to lemborexant administration was not observed according to laboratory data, and fatigue, headache, somnolence, and abnormal dreams were not observed in any patients.

## 4. Discussion

In the present study, patients who were complicated with pancreato-biliary disease were enrolled. For such cases, deep sedation is usually needed to perform endoscopic procedures. In addition, when the procedures are performed in the daytime, some patients experience insomnia after waking up from sedation, and sleeping drugs may be needed to counter this effect. Moreover, patients may sometimes experience insomnia after endoscopic procedures. In addition, the pancreato-biliary disease itself, especially pancreatic or bile duct cancer, or chronic pancreatitis due to regular alcohol use, may increase the risk of delirium because these diseases can cause insomnia, decompression, and cancer pain that requires the use of opioids. Indeed, all patients in our study had several risk factors for delirium. As delirium due to sleeping drugs can easily occur against a background of pancreato-biliary disease, careful selection of the sedative is important in these patients. Regarding risk factors associated with delirium during sleep when a sleeping drug is used, the clinical practice guidelines for adult patients in the intensive care unit do not recommend benzodiazepine administration because of the risk of delirium [9]. Another study found that benzodiazepine administration in awake patients without delirium was associated with an increased risk of delirium the next day [OR 1.04; 95% CI 1.02–1.05) [10]. Recently, a dual orexin receptor agonist such as suvorexant or Lemborexant has been available. A dual orexin receptor agonist is a novel drug developed for the treatment of insomnia that acts on orexin 1 and 2 receptors. In the orexin system that regulates the sleep–wake cycle, orexin 2 receptor is a more important receptor than orexin 1 receptor [22]. Orexin A, which is an agonist for orexin 1 receptor, is elevated in the serum of patients with agitated delirium [22]. In addition, orexin 2 receptor stimulation is associated with accelerated aggression in mice [23]. Orexin 1 receptor has a high affinity for orexin A, and orexin 2 receptor also has affinity for orexin A and B. Therefore, orexin receptor antagonists might play an important role in agitation, especially orexin 2 receptor. Suvorexnant was firstly approved as s treatment drug of insomnia. Hatta et al. conducted a multicenter, randomized, placebo-controlled clinical trial to examine whether suvorexant is effective for the prevention of delirium [24]. In this study, 230 patients were included. Compared with placebo, delirium developed significantly less often in taking suvorexant group (0% vs. 17%, respectively, *p* = 0.025). In addition, log-rank test showed that delirium developed significantly less frequency among patients who were taken suvorexant than among patients who were taken placebo (x^2^ = 06.46, *p* = 0.011). Therefore, suvorexant administration might be effective to prevent delirium. According to recent meta-analysis including 3076 patients with primary insomnia from four randomized trials, suvorexant might be associated with improvements in subjective time to sleep onset, total sleep time, and quality of sleep at 1 and 3 months. However, somnolence (RR 3.53 [95% CI, 2.19 to 5.69]), fatigue (RR 2.09 [95% CI, 1.08 to 4.03]), abnormal dreams (RR 2.08), dry mouth (RR 1.99) were frequently observed in patients receiving suvorexant compared with placebo. [25] On the other hand, because lemborexant has high affinity for orexin receptor 2, lemborexant might be also effective for insomnia without worsening delirium. Although clinical comparison study is needed, the regulation of the sleep/wake states may be achieved by orexin receptor 2 compared with orexin receptor 1 [26]. Indeed, in the present study, favorable sleep efficacy was obtained, and delirium was observed in only 1 patient despite all patients having risk factors for delirium. In addition, adverse events such as liver dysfunction was not observed in any patients according to laboratory data on next day after lemborexant administration. Moreover, fatigue, headache, somnolence, or abnormal dreams, which may be the most common adverse events of suvorexant, were not observed. Our study may be the first to evaluate the efficacy of lemborexant in patients with pancreato-biliary disease.

The present study has critical limitations. First, because the study was retrospective and all data regarding insomnia and delirium were collected by the medical records, there is a possibility of mis-collection. Therefore, the incidence of delirium may be fewer compared with prospective data collection. In addition, we could not evaluate timepoint of the assessment of delirium. This may be also influenced for decreasing the rate of delirium. Second, our study was conducted at a single center, and lacks historical controls. Therefore, a further prospective randomized controlled trial is needed to confirm the present findings.

In conclusion, lemborexant may be effective and safe for use after endoscopic procedures in pancreato-biliary patients, without increasing the risk of delirium. Further research is needed to verify our findings.

## Figures and Tables

**Table 1 jcm-12-00297-t001:** Patient characteristics.

Patients, n	64
Age, median (range), y	71 (41–90)
Sex (male: female)	47:17
Primary disease, n	
Pancreatic tumor	15
Bile duct cancer	10
Chronic pancreatitis	13
Bile duct stone	11
Hepaticojejunostomy stricture	7
Other	8
Body mass index, median, (range), kg/m^2^	21.8 (13.0–34.2)
e-GFR, median (range), mL/min	71 (6.57–174)
Mean performance status (±SD)	0.68 ± 0.88
Mean aspartate aminotransferase, (±SD, IU/L)	55.6 ± 75.8
Mean alanine transaminase, (±SD, IU/L)	56.1 ± 98.3
Mean total bilirubin, (±SD, mg/dL)	2.82 ± 5.72
Mean CRP, (±SD, mg/dL)	3.10 ± 5.18
Insomnia before hospitalization, n (%)	8 (12.5)
Regularly taking sleeping drugs, n (%)	7 (10.9)
Brotizolam	6
Triazolam	1
Lemborexant	0 (0)

**Table 2 jcm-12-00297-t002:** Risk factors for delirium.

Age > 70 years, n (%)	36 (56.3)
Dementia	10 (15.6)
Drugs, n (%)	
H2 blocker	28 (43.8)
Hypnotics	8 (12.5)
Antipsychotics	1 (1.5)
Midazolam and pentazocine	64 (100)
Regular alcohol use, n (%)	13 (20.3)

**Table 3 jcm-12-00297-t003:** Procedure outcomes.

Patients, n	64
Endoscopic procedure type	
Stone removal	11
Stent placement	33
EUS-FNA	3
EUS-FNA and biliary drainage	4
EUS-guided biliary drainage	11
EUS and biliary drainage	2
Mean dose of midazolam, (±SD, mg)	6.81 ± 2.9
Mean dose of pentazocine, (±SD, mg)	8.10 ± 3.8
Mean procedure time, (±SD, min)	36.7 ± 20.1
Adverse events during endoscopic procedures, n (%)	4 (6.3)
Hypoxemia	1
Hypotension	2
Deinhibition	1

**Table 4 jcm-12-00297-t004:** Sleep outcomes.

Dose of lemborexant, n (%) 5 mg	64 (100)
Subtype for insomnia, n (%)	
Difficulty initiating sleep	52 (81.2)
Difficulty maintaining sleep	10 (15.6)
Early-morning awakening	1 (1.6)
Non-restorative sleep	1 (1.6)
Sleep efficacy, % (n)	95.3 (61/64)
Mid-awakening after sleep, % (n)	4.9 (3/61)
Waking at the scheduled time, % (n)	98.3 (57/58)
Delirium event during the night after sleep, % (n)	1.5 (1/64)
Fall event during the night after sleep, % (n)	0 (0/64)
Mean aspartate aminotransferase, (±SD, IU/L)	64.0 ± 87.8
Mean alanine transaminase, (±SD, IU/L)	51.6 ± 70.2
Mean total bilirubin, (±SD, mg/dL)	2.9 ± 5.90
Mean CRP, (±SD, mg/dL)	2.72 ± 4.43

## Data Availability

Data is unavailable.

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
