# Peer review of "Can Lemborexant for Insomnia Prevent Delirium in High-Risk Patients with Pancreato-Biliary Disease after Endoscopic Procedures under Deep Sedation?"

_jcm, 2022, doi:10.3390/jcm12010297_

Round 1

Reviewer 1 Report

Delirium is a common complication in hospitalized patients, associated with poor negative outcomes, and interventions to reduce delirium are needed. Although this manuscript may be of interest, some issues need to be addressed. My main concern is the lack of a comparison group, which limits the ability to assess the clinical relevance of the findings. In addition, follow-up was only one night.

Introduction:

1.       The rationale behind the hypothesis that lemborexant may have a positive effect on the incidence of delirium is lacking. Reference 12 is not about lemborexant. Please add references to previous studies that have investigated the association between lemborexant and delirium. Considering that the current evidence of an association is scarce, you can also add a sentence or short paragraph about the association of suvorexant (another dual orexin receptor antagonist) and delirium.

2.       Page 1, line 44: What is the reported incidence of delirium in patients who have undergone an endoscopic procedure?

Methods:

3.       Please describe how long patients were observed for fall incidents and delirium. From the results section (lines 156-158) it seems to be only one night. If this is the case, this is a major limitation of the study, since lemborexant has a long half-life (>17 hours). Fall incidents could have occurred the next day. Moreover, it might be possible that (hyperactive) symptoms of delirium, such as agitation, were masked by lemborexant.

4.       Is it possible to include a historical group of patients with insomnia that did not receive lemborexant so you can compare the incidence of delirium?

Results:

5.       How many patients developed delirium after the first night? Is it possible that symptoms of delirium were masked during the first night?

6.       And how many patients continued to use lemborexant after the first night?

Discussion:

7.       Line 203: Orexin A is an agonist, not an antagonist.

Author Response

Response to Reviewer 1 Comments

Point 1:    The rationale behind the hypothesis that lemborexant may have a positive effect on the incidence of delirium is lacking. Reference 12 is not about lemborexant. Please add references to previous studies that have investigated the association between lemborexant and delirium. Considering that the current evidence of an association is scarce, you can also add a sentence or short paragraph about the association of suvorexant (another dual orexin receptor antagonist) and delirium.

Response 1: Thank you for your comment and suggestion. I comletely agree with your opinion. Also, we are sorry for this mistake. We corrected references, and added a sentence regarding suvorexant.

Point 2: Page 1, line 44: What is the reported incidence of delirium in patients who have undergone an endoscopic procedure?

Response 2: We are sorry for this insufficient description. During pancreatobiliary endoscopic procedure, deep sedation is required. During this procedure, midazolam is usually used as sedation, therefore, pancreato-biliary patients who undergo endoscopic procedures has potential risk of delirium. We corrected this sentence. Once again, thank you for your comment.

Point 3: Please describe how long patients were observed for fall incidents and delirium. From the results section (lines 156-158) it seems to be only one night. If this is the case, this is a major limitation of the study, since lemborexant has a long half-life (>17 hours). Fall incidents could have occurred the next day. Moreover, it might be possible that (hyperactive) symptoms of delirium, such as agitation, were masked by lemborexant.

Response 3: We are sorry for insufficient description. Delirium was checked during 72 hours after lemborexant administration. We corrected definitions and result section. 

Point 4: Is it possible to include a historical group of patients with insomnia that did not receive lemborexant so you can compare the incidence of delirium?

Response 4: We completely agree with your opinion. If a histrical control group is added, our study may be improved. However, as mentioned in our TEXT, our data was retrospectively collected. This was one of limitation of our study. If histrical control group was added, data may not be more reliability because of retrospetive nature. In addition, our study has been already approved in our hospital, therefore, protocol change cnould not be allowed in this stage. However, there was no evidence regarding the efficacy of lemborexant in patients with pancreato-biliary disease. Therefore, we believe our study may be landmark. I hop you understand this explanation.

Point 5: How many patients developed delirium after the first night? Is it possible that symptoms of delirium were masked during the first night?

Response 5: As mentioned above, the frequency of delirium was evaluated during 72 hours. During this time, delirum was observed in only one patient after the first night. We added this in the result section.

Point 6: And how many patients continued to use lemborexant after the first night?

Response 6: In this study, there were no patient who continued to use lembprexanat after the first night among 72 hours. Thank you for your comment.

Point 7: Line 203: Orexin A is an agonist, not an antagonist.

Response 7: We deeply appologized for this mistake. We corrected this.

           Finally, thank you for your valuable comments and suggestions.

Reviewer 2 Report

Thank you very much for this interesting pilot study. The neccessity of such an evaluation for the patient sample, that was examined, is clearly explained.

In general the wording needs some revision. The grammar of the first sentence of the introduction should be revised. There is also some wording that is difficult to understand in lines 156 to 158.

To assess the incidence of delirium retrospectively is often difficult as in hospital routine delirium is often underrecognised. Therefore, the results regarding the incidence of delirium should be discussed. As delirium is a second outcome it would be interesting to know how the assessment was conducted. For example the timepoint of the assessment could be an influencing factor.

It would be interesting to know how was the alcohol consumption assessed, as this is mentioned as a risk factor for delirium. There should be a rational provided that explains why certain laboratory values were examined and how they might affect the primary outcome. It would be interesting to know how the quality of sleep was assessed. The description of surgery is too extended as the way the surgery was conducted is not subject to the article. It would be of more interest to explain how the safety of lemborexant was assessed.

The structure of the article should be changed. The first part of the discussion (lines 163 to 208) is better placed in the introduction as it provides general information about delirium, midazolam and lemborexant. Therefore, the discussion seems to be too short. It would be better to discuss the results in the context of the characteristics of the patient sample and the safety of lemborexant.

Author Response

Response to Reviewer 2 Comments

Point 1: In general the wording needs some revision. The grammar of the first sentence of the introduction should be revised. There is also some wording that is difficult to understand in lines 156 to 158.

Response 1: We are sorry for these mistakes. We carefully corrected. Thank you for your comments.

Point 2: To assess the incidence of delirium retrospectively is often difficult as in hospital routine delirium is often underrecognised. Therefore, the results regarding the incidence of delirium should be discussed. As delirium is a second outcome it would be interesting to know how the assessment was conducted. For example the timepoint of the assessment could be an influencing factor.

Response 2: Thank you for your suggestion. As you pointed out, our study has critical limitation such as retrospective nature. Although this has been already described as limitation, this limitation was more clearly described. Also, we agree with your suggestion. Timepoint of the assesment could be an influencing factor. However, as mentioned before sentence, correct data collection may be challenging because of retrospective nature. We plan to conduct prospetive study in near future. This also was added as limitation. I hope understand this explanation.

Point 3: It would be interesting to know how was the alcohol consumption assessed, as this is mentioned as a risk factor for delirium. There should be a rational provided that explains why certain laboratory values were examined and how they might affect the primary outcome. It would be interesting to know how the quality of sleep was assessed. The description of surgery is too extended as the way the surgery was conducted is not subject to the article. It would be of more interest to explain how the safety of lemborexant was assessed.

Response 3: We also agree with your suggestion. Alcohol may be important to discuss about delirium. However, data was collected using by medical record, certain amount of alcohol could not be described. However, in our hospital, Regular alcohol use was defined as a daily intake of >60 g of ethanol. This was described in the method section.

Laboratory data was shown to evaluate liver damage due to lemboxant and inflammatory data may be infulenced for the frequency of delirium. We added several sentences in the discussion section. Also, we deleted several sentences about endoscopic procedrues.

Point 4: The structure of the article should be changed. The first part of the discussion (lines 163 to 208) is better placed in the introduction as it provides general information about delirium, midazolam and lemborexant. Therefore, the discussion seems to be too short. It would be better to discuss the results in the context of the characteristics of the patient sample and the safety of lemborexant.

Response 4: We strongly agree with your suggestion. We corrected, accordingly. Once again, thank you for your suggestions.

Round 2

Reviewer 2 Report

Dear Sirs,

I revised the changes you made to your interesting paper. Still I think there are some minor points, that should be clarified further:

1. you should mention the general assessment of alcohol consumption- especially the duration of consumption- in the limitations, as you found an association of alcohol consumption and postoperative delirium. This association might  not be true in patient samples with different patterns of alcohol consumption.

2. You mentionend additional costs caused by postoperative delirium. It would be interesting to know for which countries those figures apply.

3. Still there are some minor mistakes in the English wording in lines 61-62, 84-86, 94-96, 228-231, 271-272, 300-302, and 314-315. These sentences should be corrected.

Thank you.

Author Response

Response to Reviewer 2 Comments

Point 1: you should mention the general assessment of alcohol consumption- especially the duration of consumption- in the limitations, as you found an association of alcohol consumption and postoperative delirium. This association might  not be true in patient samples with different patterns of alcohol consumption..

Response 1: We strongly agree with your opinion. As you mentioned, because our study was retrospective, therefore, we could not evaluate certain amont of alcohol consumption. We added this sentence as limitation. Thank you very much for your valuable opinion.

Point 2: You mentionend additional costs caused by postoperative delirium. It would be interesting to know for which countries those figures apply.

Response 2: This was based on U.S. We added this in the TEXT.

Point 3: Still there are some minor mistakes in the English wording in lines 61-62, 84-86, 94-96, 228-231, 271-272, 300-302, and 314-315. These sentences should be corrected..

Response 3: We are sorry for these mistakes. We corrected, accordingly. Once again, thank you for your valuable opinions.